# Stepwise heating in Stille polycondensation toward no batch-to-batch variations in polymer solar cell performance

Sang Myeon Lee[1], Kwang Hyun Park[1], Seungon Jung[1], Hyesung Park[1] & Changduk Yang [1]

For a given π-conjugated polymer, the batch-to-batch variations in molecular weight ($M_w$) and polydispersity index (Đ) can lead to inconsistent process-dependent material properties and consequent performance variations in the device application. Using a stepwise-heating protocol in the Stille polycondensation in conjunction with optimized processing, we obtained an ultrahigh-quality PTB7 polymer having high $M_w$ and very narrow Đ. The resulting ultrahigh-quality polymer-based solar cells demonstrate up to 9.97% power conversion efficiencies (PCEs), which is over 24% enhancement from the control devices fabricated with commercially available PTB7. Moreover, we observe almost negligible batch-to-batch variations in the overall PCE values from ultrahigh-quality polymer-based devices. The proposed stepwise polymerization demonstrates a facile and effective strategy for synthesizing high-quality semiconducting polymers that can significantly improve device yield in polymer-based solar cells, an important factor for the commercialization of organic solar cells, by mitigating device-to-device variations.

[1] Department of Energy Engineering, School of Energy and Chemical Engineering, Low Dimensional Carbon Materials Center, Perovtronics Research Center, Ulsan National Institute of Science and Technology (UNIST), 50 UNIST-gil, Ulju-gun, Ulsan 44919, Republic of Korea. Correspondence and requests for materials should be addressed to H.P. (email: hspark@unist.ac.kr) or to C.Y. (email: yang@unist.ac.kr)

π -Conjugated polymers continue to revolutionize the field of printed and flexible organic electronics[1–4]. Pertinent to synthesis, versatile organic building blocks have been widely coupled to form $sp^2$–$sp^2$ carbon bonds with polyaromatic conjugations via various cross-coupling reactions[5–9]. With a large number of functional groups available under less harsh reaction conditions along with high yields, palladium-catalyzed Stille polycondensation has been extensively employed for construction of various π-conjugated donor–acceptor (D–A) polymers[10–12]. This has led to the advanced technologies for numerous applications, ranging from polymer light-emitting diodes[13], polymer field-effect transistors[14], and polymer solar cells (PSCs)[15]. Owing to the current impetus in advancing the field of energy-pertinent polymer applications, many chemists have recently developed state-of-the-art polymers for bulk-heterojunction (BHJ) PSCs, achieving power conversion efficiencies (PCEs) exceeding 10%[16–19]. Despite such successful accomplishments, they have a ubiquitous experimental limitation. Unlike the well-defined structures of small molecular semiconductors, for a given polymer prepared from transition metal-catalyzed cross-coupling reactions, the batch-to-batch variations in molecular weight ($M_w$) and polydispersity index (Đ) have given rise to inconsistent process-dependent material properties and consequent performance variations in device applications[20–24]. Therefore, optimized synthetic methodologies have been rigorously investigated to improve Stille polycondensation reaction protocols, e.g., through examining the role of the catalyst, solvent, and other reaction conditions[10,25,26]. Nonetheless, to date, such attempts have not surmounted the aforementioned batch-to-batch problems. Therefore, a formidable challenge in π-conjugated polymer chemistry is to develop a robust synthetic methodology toward higher-quality polymeric precision with high $M_w$ and narrow Đ, thus overcoming the current limitation of the statistically determined nature of synthetic polymers.

Here, we report an efficient and facile stepwise Stille polycondensation with an additionally embedded low-temperature step for the preferential step growth of low-$M_w$ fractions to formulate high-quality polymers. We choose a well-known high-performance polymer, poly[4,8-bis[(2-ethylhexyl)oxy]benzo[1,2-b:4,5-b′]dithiophene-2,6-diyl-alt-3-fluoro-2-[(2-ethylhexyl)carbonyl]thieno[3,4-b]thiophene-4,6-diyl] (PTB7)[27], as a test-bed polymeric material for our study (Fig. 1). After extensive screenings of the key aspects of the polymerization in context, we employ stepwise Stille polycondensation upon the optimized synthesis condition, affording ultrahigh-quality PTB7 having high $M_w$ (223 kDa) and very narrow Đ (1.21). Despite the similar optical, electrochemical, and morphological properties between the stepwise protocol-derived and commercially available PTB7 polymers, the resulting ultrahigh-quality polymer-based solar cells exhibit superior PCEs of up to 9.97% with impressively negligible device-to-device performance variations, constituting a

substantial improvement from control devices based on the commercial polymer. The stepwise Stille polycondensation process is readily adoptable to a wide range of general π-conjugated polymers requiring ultrahigh quality and can be immediately incorporated into synthetic organic chemists' repertoire as a principal reaction protocol.

## Results

**Conventional Stille polymerization.** High purities in a pair of D- and A-monomers play pivotal roles for a precise stoichiometric balance in polymer chemistry, and even marginal imbalances in stoichiometry can have a significant impact on the degree of polymerization[20,28,29]. Therefore, to minimize the possibility of stoichiometric imbalances from the impurity of the reactant monomers, we further refined two purified monomers (**1** and **2**) using a JAI-LC9160II preparative gel permeation chromatograph (GPC) with a reversed-phase column in acetonitrile:tetrahydrofuran (THF) (6:4 vol%). We verified the resulting high purity by elemental analysis and high-resolution mass spectrometry (HRMS; Supplementary Fig. 1 and Table 1). Previous work[27,30] reports the Stille polycondensation of PTB7 using Pd(PPh$_3$)$_4$ (4.0 mol%) in toluene:N, N-dimethylformamide (DMF) (4:1 vol%, 0.10 M) at 120 °C under argon for 12 h, providing a high-performance polymeric material in PSCs. Therefore, we first performed Stille polycondensation under the same conditions, which allowed us to obtain PTB7 with an acceptable $M_w$ of 70.6 kDa and a Đ of 2.18 in 76% yield (entry 1, Table 1). We evaluated the $M_w$ and Đ values by high-temperature GPC at 120 °C using 1,2,4-tricholorobenzene as the eluent.

To determine the optimal combinatorial settings for a further improved Stille polycondensation condition of PTB7, we examined the overall synthesis scope and optimized the reaction conditions with respect to parameter control such as the catalyst, reaction time, temperature, solvent, and concentration. Initially, we attempted to utilize a more air-stable Pd$_2$(dba)$_3$(2.0 mol%):P(o-tolyl)$_3$ (8.0 mol%) precatalytic system compared to the aforementioned Pd(PPh$_3$)$_4$-based scheme, which led to a slightly lower $M_w$ of 61.2 kDa in 55% yield. Considering the $M_w$ and yield of the polymer, we thus adopted Pd(PPh$_3$)$_4$ as the optimal catalyst to examine the following polymerization processes.

For prolonged reaction times of 24 and 36 h (entries 3 and 4, respectively), the corresponding $M_w$ values (70.9 to 72.7 kDa) were similar to that obtained from the analogous reaction conducted over the course of 12 h, suggesting that the polycondensation reaction proceeded to completion in <12 h under the specific test conditions. Although employing different temperatures (100 and 140 °C) offered narrower Đ values of 1.70 to 1.90, there was a trade-off of lowered $M_w$s. The evaluated temperature of 140 °C resulted in a yield as high as 82%, which is twice that at 100 °C. In addition, we observed a sharp decline of

**Fig. 1** Synthetic pathways for the PTB7 test-bed polymer. **a** Conventional Stille reaction is comprised of only steady heating procedure at 120 °C. **b** Stepwise Stille polycondensation in this work adopted cooling step at 60 °C after which initial heating was applied at 120 °C for first 1 h

**Table 1 Conventional Stille polymerization of PTB7 under screening reaction parameters**

| Entry | Solvent (vol%) | Temp. (°C) | Time (h) | $M_n$ (kDa)[a] | $M_w$ (kDa)[a] | $Đ$[a] | Yield (%)[b] |
|---|---|---|---|---|---|---|---|
| 1 | Toluene:DMF (4:1) | 120 | 12 | 32.3 | 70.6 | 2.18 | 76 |
| 2[c] | Toluene:DMF (4:1) | 120 | 12 | 26.9 | 61.2 | 2.28 | 55 |
| 3 | Toluene:DMF (4:1) | 120 | 24 | 34.1 | 70.9 | 2.08 | 62 |
| 4 | Toluene:DMF (4:1) | 120 | 36 | 32.6 | 72.7 | 2.23 | 71 |
| 5 | Toluene:DMF (4:1) | 100 | 36 | 25.1 | 42.8 | 1.70 | 43 |
| 6 | Toluene:DMF (4:1) | 140 | 12 | 31.5 | 59.7 | 1.90 | 82 |
| 7 | Toluene | 120 | 36 | 17.1 | 28.4 | 1.66 | 37 |
| 8 | Toluene:DMF (1:1) | 120 | 12 | 16.3 | 46.4 | 2.84 | 33 |
| 9[d] | Toluene:DMF (4:1) | 120 | 12 | 48.3 | 94.2 | 1.95 | 75 |
| 10 | Toluene:DMF (4:1) | 120 | 1 | 12.7 | 40.0 | 3.15 | –[e] |
|  |  |  | 6 | 20.9 | 60.9 | 2.91 | –[e] |

The conventional Stille polycondensation was carried out under an argon atmosphere in a long Schlenk tube of monomers **1** and **2** in 0.10 M solution, and 4.0 mol% of Pd(PPh$_3$)$_4$
[a] Number-average ($M_n$), weight-average ($M_w$) molecular weights, and $Đ$ values were determined from GPC measurement using 1,2,4-trichlorobenzene at 120 °C calibrated with polystyrene as standard
[b] Yields were estimated from the amounts of the chloroform fractions
[c] Different catalyst system, Pd$_2$(dba)$_3$ (2.0 mol%):P(o-tolyl)$_3$ (8.0 mol%) was adopted
[d] The concentration of solutions was lowered to 0.05 M
[e] Each fraction was extracted by a syringe, then precipitated in methanol with Soxhlet purification for only GPC analysis

**Table 2 Stepwise Stille polymerization with the optimization data**

| Entry | Solvent (Conc.) | Catalyst (mol%) | $M_n$ (kDa)[a] | $M_w$ (kDa)[a] | $Đ$[a] | Yield (%)[b] |
|---|---|---|---|---|---|---|
| 11 | Toluene:DMF (0.05 M) | Pd(PPh$_3$)$_4$ (4.0) | 67.2 | 90.1 | 1.34 | –[c] |
| 12[d] | Toluene:DMF (0.05 M) | Pd(PPh$_3$)$_4$ (4.0) | 43.9 | 82.6 | 1.88 | –[c] |
| 13[d] | Toluene:DMF (0.05 M) | Pd(PPh$_3$)$_4$ (4.0) | 42.5 | 99.8 | 2.35 | –[c] |
| 14 | Toluene:DMF (0.10 M) | Pd(PPh$_3$)$_4$ (4.0) | 31.3 | 65.1 | 2.08 | 63 |
| 15 | Toluene:DMF (0.026 M) | Pd(PPh$_3$)$_4$ (4.0) | 82.1 | 125 | 1.52 | 65 |
| 16 | Toluene:DMF (0.026 M) | Pd(PPh$_3$)$_4$ (3.0) | 110 | 151 | 1.37 | 81 |
| 17 | Toluene:DMF (0.026 M) | Pd(PPh$_3$)$_4$ (2.0) | 151 | 195 | 1.29 | 79 |
| 18 | Toluene:DMF (0.026 M) | Pd(PPh$_3$)$_4$ (1.0) | 184 | 223 | 1.21 | 85 |
| 19[e] | Conventional polymerization | Pd(PPh$_3$)$_4$ (1.0) | 46.9 | 75.8 | 1.62 | 72 |

The stepwise Stille polycondensation was carried out under an argon atmosphere in a long Schlenk tube of monomer **1**, **2**, and Pd(PPh$_3$)$_4$, and procedures included the initial heating at 120 °C for 1 h, the cooling step at 60 °C for 11 h, and the final heating at 120 °C for 1 day
[a] $M_n$, $M_w$, and $Đ$ values were determined from GPC measurement using 1,2,4-trichlorobenzene at 120 °C calibrated with polystyrene as standard
[b] Yields were estimated from the amounts of the chloroform fractions
[c] Each fraction was extracted by a syringe, then precipitated in methanol with Soxhlet purification for only GPC analysis
[d] The cooling temperature was investigated at 80 and 100 °C for entries 12 and 13, respectively
[e] 1.0 mol% Pd catalyst was employed for conventional Stille polymerization. A mixture of monomer **1**, **2**, and Pd(PPh$_3$)$_4$ in a binary solvent of toluene and DMF (4:1 vol%, 0.026 M) was reacted at 120 °C for 1 day in a long Schlenk tube under an argon condition

$M_w$ to 28.4 kDa with only toluene solvent (entry 7). This indicates that the toluene:DMF system can improve the solubility of polymers with the catalyst-stabilizing effects, leading to excellent yields of high-$M_w$ polymers. Such benefits induced by mixed solvent systems have been previously reported[25,31,32]. However, a higher DMF-to-toluene ratio (1:1 vol%) severely lowered the $M_w$, which may be attributed to the insufficient solubility of the growing polymer chain in the mixed ratio despite more pronounced catalyst stabilization. With an optimal toluene:DMF (4:1 vol%) system, the polymerization employing a lower monomer concentration (down to 0.05 M) was more effective in generating a high yield of higher-$M_w$ polymers (94.2 kDa) having narrow $Đ$ (1.95). Table 1 summarizes the aforementioned polymerization conditions and relevant data for the PTB7 polymer.

**Stepwise Stille polymerization**. We speculate that various components with different $M_w$s, formed during the initial stages of polymerization, lead to broad $M_w$ distributions when the duration of the step-growth polymerization is extended. The $Đ$ data of the polymers synthesized under the short reaction times (entry 10 and Supplementary Fig. 3a) corroborates this speculation. In principle, the different $M_w$ components undergo different kinetics and thermodynamics driven toward coupling reactions.

Therefore, we assume that their reaction selectivity can be controlled by the reaction temperature, and the temperature-dependent $Đ$ values (see entries 5 and 6) validate this assumption. Therefore, we further investigated precise temperature-controlled Stille polycondensations in combination with stepwise-heating protocols to obtain a higher quality of PTB7 polymer under the optimized solvent (toluene:DMF, 4:1 vol%), catalyst (Pd(PPh$_3$)$_4$, 4.0 mol%), and concentration (0.05 M) experimental conditions.

We first performed polymerization at the optimal temperature (120 °C) for 1 h and then cooled the reaction mixture to 60 °C for 11 h as a selectivity control step. Over this time period, the low-$M_w$ species would react preferentially, and thus, the window of the $M_w$ distributions can potentially be reduced (Supplementary Fig. 3b). It is noteworthy that all of the polymeric fractions formed at the low-temperature step (60 °C) are readily dissolved in the mixed reaction solvent (Supplementary Fig. 3c), suggesting that the reactivity variation caused by their solubility issues is negligible. After the cooling stage, we reverted the reaction temperature to the initial setting for finalizing the step-growth reaction. Such a stepwise polymerization approach yielded higher-$M_w$ polymeric fractions ($M_w$ greater than 90 kDa) having a narrower $Đ$ of 1.34 (entry 11, Table 2). High $M_w$ values of the PTB7 polymer could also be obtained through the temperature changes (80 and 100 °C) from the preference-control step, but

this approach adversely affected the $Đ$ values (entries 12 and 13). Therefore, we further investigated the effect of changes in the concentration and catalyst quantities at a fixed temperature of 60 °C over the course of the low-temperature maturation process (entries 14 to 18).

A comparison of the entries' results tested for various concentrations and catalysts indicates that in addition to the notably high $M_w$s observed (exceeding 125 kDa at 0.026 M), $Đ$ values gradually reduced in accordance with the decreasing catalyst loading. This is probably attributable to the suppression of multiple reaction pathways at low catalyst loading[6,25,33,34]. Through further optimization processes, we obtained the best-quality PTB7 polymer having $M_w$ of 223 kDa and $Đ$ of 1.21 using a combination of 0.026 M concentration and a catalyst concentration of 1.0 mol%. For a fair comparison, the conventional Stille polycondensation was also performed using the optimized condition above, producing PTB7 with only moderate $M_w$ (entry 19), which clearly corroborates the efficacy of stepwise polymerization to achieve a higher-quality PTB7 with high $M_w$ and narrow $Đ$. Supplementary Table 4 lists the remainder of the conditions tested over the course of our study.

**Characterizations of PTB7 polymers.** Several reports have demonstrated the influence of polymer $M_w$s on various optical, physicochemical, and morphological properties[35–40]. In this regard, we explored the optical and electrochemical properties, morphology, and molecular organization for two representative batches (entries 17 and 18) using ultraviolet (UV)–visible (Vis) absorption spectroscopy, cyclic voltammetry, atomic force microscopy (AFM), transmission electron microscopy (TEM), and grazing-incidence wide-angle X-ray diffraction (GIWAXD), summarized in the Supplementary Table 4 and Table 5. The film absorption coefficients ($\varepsilon$) varied from 93,900 to 133,000 cm$^{-1}$ for entries 17 and 18, implying that the increase of $M_w$s can result in the enhanced light-absorbing capability. Nonetheless, the two test samples exhibited similar physical features (nearly identical absorption shape, optical bandgap of 1.67 eV, HOMO:LUMO of

−5.34:−3.61 eV, small surface roughness of 1.0 nm, lamellar distance of 19.5 Å, and π−π stacking distance of 3.9 Å), as revealed by a series of the aforementioned analysis (Fig. 2), which to some extent may reflect the sufficiently high quality that correlates with nearly identical structural aspects.

Next, we further compared and quantified the difference between the resulting polymers prepared from our optimized stepwise and conventional protocols (entries 17, 18, and 19) and commercially available PTB7 products, which we purchased from Derthon and 1-Material companies. As shown in Fig. 3, both entries 17 and 18 show significantly higher $M_w$ and narrower $Đ$ values than those of the entry 19 and commercial products, which we determined by the same processing GPC analysis (Supplementary Table 7). Note that the different commercial products used in this study are denoted as Commer-1 and Commer-2, respectively. These results clearly indicate that stepwise polymerization is an effective strategy to achieve a narrow distribution of high-$M_w$ polymers.

## Discussion

To assess the effect of polymer quality control on photovoltaic properties, we fabricated archetypal single junction PSC devices of indium-tin oxide (ITO)/poly(3,4-ethylenedioxythiophene): poly(styrenesulfonate) (PEDOT:PSS)/polymer:[6,6]-phenyl-C$_{71}$-butyric acid methylester (PC$_{71}$BM)/Al based on four polymers (entries 17, 18, 19, and Commer-1). In accordance with the aforementioned optimized processing conditions, we spin-coated the PTB7:PC$_{71}$BM blend (1:1.7 wt%), 12 mg mL$^{-1}$ concentration, in a mixed solvent of chlorobenzene (CB):1,8-diiodooctane (DIO) (97:3 vol%). We chose Commer-1 as a commercial reference sample because of its higher quality from a materials standpoint and better PSC characteristics in the initial screening test relative to that of Commer-2 (Supplementary Fig. 6 and Table 8). The result of entry 19 obtained from conventional Stille polymerization was also included in the evaluation of the PSC performance for the sake of comparison.

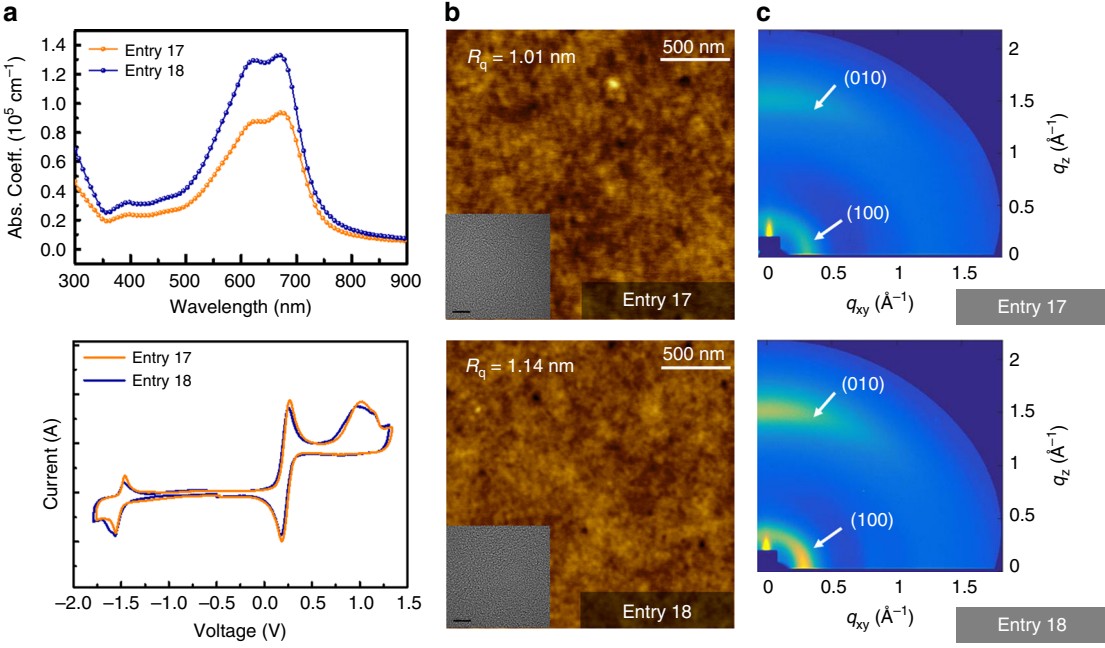

**Fig. 2** Optical, electrochemical, and morphological properties of entries 17 and 18. **a** UV–Vis absorption spectra and cyclic voltammograms. **b** AFM height images in a scale of 500 nm with inset TEM images (5 nm scale). **c** GIWAXD images of in-plane and out-of-plane

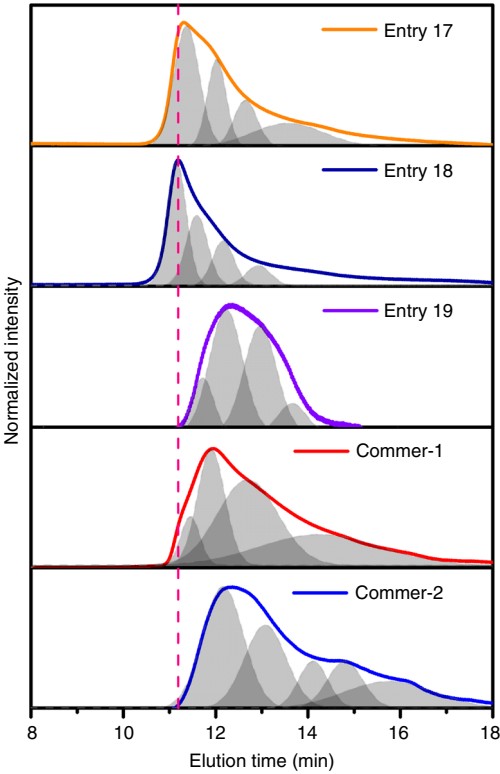

**Fig. 3** GPC data for test batches and commercial polymers. The multiple curves were deconvoluted through the fitting of each GPC curve

Figure 4a shows the current density−voltage ($J$–$V$) curves measured under the simulated AM 1.5 G at 100 mW cm$^{-2}$ irradiance, and Table 3 shows the corresponding key photovoltaic parameters. The Commer-1:PC$_{71}$BM control device showed a short-circuit current density ($J_{SC}$) of 16.5 mA cm$^{-2}$, an open-circuit voltage ($V_{OC}$) of 0.72 V, and a fill factor ($FF$) of 61%. This indicates a PCE of 8.02% (PCE$_{avg}$ = 7.26%), which is similar to the results previously reported from the highly optimized PTB7-based devices[41–45]. The device performance for the entry 19 was lowest with a PCE of 6.98% (PCE$_{avg}$ = 5.92%). On the other hand, both entries 17- and 18-based devices display significant improvements in the $J_{SC}$s, having similar $V_{OC}$s of 0.71 V and $FF$s of 62%. The increase in $J_{SC}$ values is in good agreement with the results of external quantum efficiency (EQE) measurements from the corresponding devices (Fig. 4b). In addition, the trend of the EQE enhancements is well-matched with the increased absorption spectra of the blend films (Supplementary Fig. 7), which is consistent with what was previously reported in that the photo-current correspondingly increases with $M_w$ values[46,47]. Consequently, we achieved a PCE of up to 9.97% (PCE$_{avg}$ = 9.69%) in the best-quality entry 18-based device, which, our best knowledge, is the highest value reported for PTB7-based PSCs. More importantly, the entry 18-based device shows superior reproducibility in regard to all photovoltaic parameters; more than 30 devices lie within trivial error ranges of 5.88% (to 95% confidence level), whereas the control devices (entry 19 and Commer-1) show large performance variations in each device type (Fig. 4c). The origin of improvements in both the PCEs and reproducibility of PSCs based on entry 18 is consistent with what was previously suggested in that a lower density of radical-type paramagnetic defects in PTB7 with high $M_w$ leads to improved device functionality[48]. These observations imply that polymer chain quality control is an important contributor to resolving the challenging issue of device-to-device variations in PSCs.

Moreover, to explore the versatility of the stepwise-heating protocol, we attempted to prepare various other polymers, including thienopyrroledione-, diketopyrrolopyrrole-, quinoxa-line-, and thiophene-based monomeric systems by using con-ventional- and stepwise-heating systems, respectively (Supplementary Fig. 8 and Note 1). As can be seen in the relevant GPC results (Supplementary Fig. 9 and Table 9), in all cases, the stepwise polymerization produced the polymeric components with higher $M_w$ and narrower Đ values compared to those of the corresponding ones obtained from the conventional method. The PSCs based on the resulting polymers were comparatively investigated in the same device architecture as described above (Supplementary Note 2, details of the optimized conditions for each polymer). The devices based on the stepwise batches showed higher PCEs than those of the corresponding ones fabricated from the conventional method (Supplementary Fig. 10). Note that the higher-quality poly[4,8-bis[(2-ethylhexyl)oxy]benzo[1,2-$b$:4,5-$b'$]dithiophene-2,6-diyl-$alt$-5-octylthieno[3,4-$c$]pyrrole-4,6-dione-1,3-diyl] (PBDT-TPD) prepared from the stepwise tool exhibited a slightly lower PCE than that of the sample with conventional tool, due to its poorer film formation with aggres-sive aggregation behavior induced by the low solubility and processability upon the device fabrication[49,50]. These overall results propose the feasibility of stepwise Stille polymerization for a broader application.

In summary, we introduced the stepwise-heating protocol for Stille polycondensation, applied it to the preferential step growth of low-$M_w$ fractions, and successfully demonstrated our protocol for the PTB7 system. We optimized the process conditions by thoroughly screening a comprehensive set of control parameters, including catalyst, reaction time, temperature, solvent, and con-centration. Compared to the commercial PTB7 product-based PSCs, the ultrahigh-quality PTB7 (high $M_w$ and narrow Đ) pre-pared via optimized stepwise polymerization shows substantial improvement of the $J_{SC}$ exceeding 20 mA cm$^{-2}$. This results in an outstanding PCE of up to 9.97% (PCE$_{avg}$ = 9.69%), which is by far the highest value reported for PTB7-based PSCs. Moreover, the quality-controlled PTB7 can significantly suppress the batch-to-batch variations in solar cell performance, leading to excellent reproducibility in the overall PCE values. Our findings will not only provide important progress for prevalent semiconducting polymer reactions but also help overcome the inveterate dis-advantage of polymer-based electronics applications.

## Methods

**Materials and monomer synthesis**. All the chemicals including solvents, reagents, and catalysts for this work, were purchased from Sigma-Aldrich, Alfa Aesar chemical company, Tokyo Chemical Industry Co., Ltd., and Derthon Optoelectronic Materials Science Technology Co., Ltd. Unless otherwise specified, such chemicals were used without any further purification. (*Caution*: Trimethyltin chloride and other organotin-related compounds are highly toxic to cause irritation and burns of the skin and eyes with fatal damage on the central nervous system. High and repeated exposure can cause loss of hearing, weakness, confusion, and sei-zures[51,52].) 2,6-Bis(trimethyltin)-4,8-bis(2-ethylhexyloxy)benzo[1,2-$b$:4,5-$b'$] dithiophene was synthesized according to the literature[30] and 2-ethylhexyl-4,6-dibromothieno[3,4-$b$]thiophene-2-carboxylate was purchased from Derthon, both of which were purified with recrystallization and chromatography using a reverse-phase column, JAIGEL-ODS-AP, SP-120–10 (20 × 250 mm , i.d., Japan Analytical Industry Co., Ltd.) in a mixture of acetonitrile and THF as the mobile phase. The elution was monitored at 256 nm with a UV detector.

**Conventional Stille polycondensation**. We dissolved monomers **1** (100 mg, 0.13 mmol) and **2** (61.1 mg, 0.13 mmol) in a single solvent or a mixture of toluene and DMF (0.10 and 0.05 M) in a long Schlenk tube. After intensive bubbling with argon for 20 min, for accuracy, we injected the solution of Pd catalyst in toluene (5.2 mM, 4.0 mol%) with subsequent purging for 10 min. We kept the mixture in a preheated oil bath at certain temperatures for various durations. Table 1 summarizes the pertinent conditions. After cooling to room temperature, we purified the pre-cipitates in methanol by Soxhlet extraction with methanol, acetone, hexane, and chloroform. We quantitated the $M_n$, $M_w$, and Đ values by high-temperature GPC

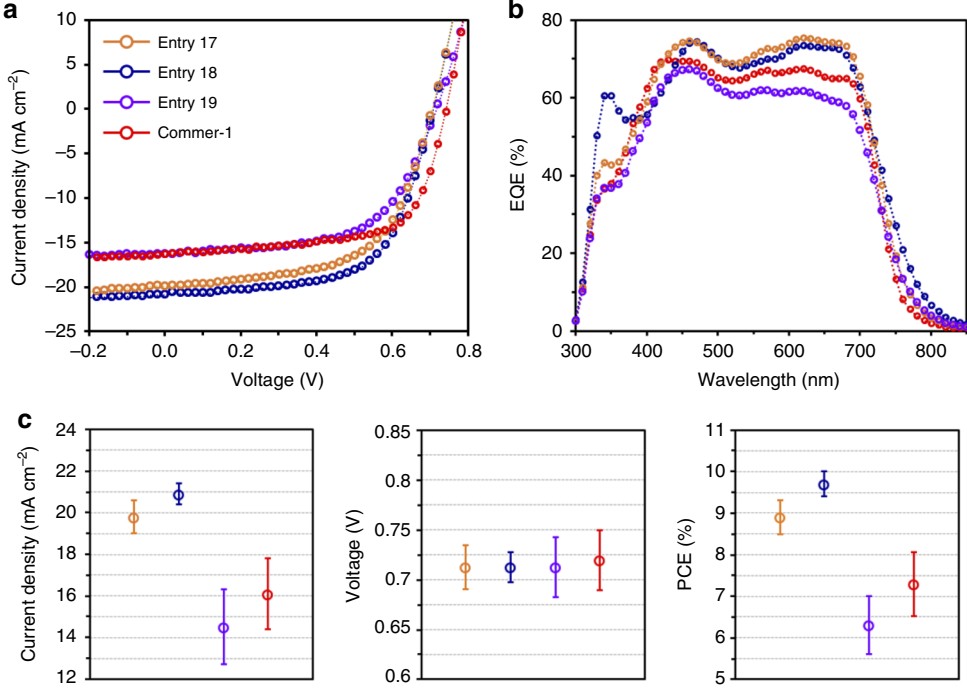

**Fig. 4** Photovoltaic performance from the selected entries and Commer-1. **a** J–V characteristics for BHJ solar cells. **b** EQE spectra in a range from 300 to 850 nm. **c** Variations in device parameters of $J_{SC}$s, $V_{OC}$s, and PCEs. All error bars with average values were obtained from more than 30 devices for each polymer

### Table 3 Device parameters for entries 17, 18, 19, and Commer-1

| Samples | $J_{SC}$ (mA cm$^{-2}$) | $V_{OC}$ (V) | FF (%) | PCE (%) max/avg.[a] |
|---|---|---|---|---|
| Entry 17[b] | 19.8 ± 0.4 | 0.71 ± 0.02 | 62 ± 2 | 9.35/8.93[a] |
| Entry 18[b] | 20.9 ± 0.3 | 0.71 ± 0.01 | 62 ± 2 | 9.97/9.69[a] |
| Entry 19[b] | 16.3 ± 0.6 | 0.72 ± 0.03 | 60 ± 3 | 6.98/5.92[a] |
| Commer-1[b] | 16.5 ± 0.5 | 0.72 ± 0.02 | 61 ± 1 | 8.02/7.26[a] |

*avg.*average
[a] The average values were obtained from over 30 devices with the standard deviation
[b] The thicknesses of the polymer blending films with PC$_{71}$BM are 85, 80, 78, and 83 nm, respectively

and estimated the yields from the isolated fraction in chloroform. The additional screenings in the conventional reactions were performed with changes in the method (see the Supplementary Table 3).

**Stepwise Stille polycondensation.** We dissolved monomers **1** (100 mg, 0.13 mmol) and **2** (61.1 mg, 0.13 mmol) in a binary mixture of toluene and DMF (0.10 to 0.026 M) as a 4:1 ratio in a long Schlenk tube[53]. After intensive bubbling with argon for 20 min, we injected the solution of Pd catalyst in toluene (5.2 to 1.3 mM) with subsequent purging for 10 min. We reacted the mixture in a preheated oil bath at 120 °C for 1 h for the initial stage, after which allowed the temperature to reduce and kept the tube cooled for 11 h. We then reset the temperature to 120 °C and continuously maintained this temperature for 1 day. We extracted small quantities of the mixture for GPC measurement with a syringe before subsequent heating and then performed rapid precipitation in methanol as well as Soxhlet purification. We confirmed the high-temperature $^1$H nuclear magnetic resonance (NMR) results (Supplementary Fig. 11 of representative entries 17 and 18 and Commer-1).

**General characterization.** Elementary analyses were performed by a Flash 2000 element analyzer (Thermo Scientific). HRMS spectra were measured using a Q Exactive™ Plus Hybrid Quadrupole Orbitrap™ mass spectrometer (Thermo Scientific). The polymers used for comparison in this work were purchased from Derthon and 1-Material and the characterization data for polymeric parameters are summarized in Supplementary Table 7. $^1$H NMR spectra were recorded on a Varian VNRS 600 MHz spectrometer using deuterated 1,1,2,2-tetrachloroethane ($C_2D_2Cl_4$) as solvent at 348 K. UV–Vis–NIR absorption spectra were obtained using a UV-1800 (SHIMADZU) spectrophotometer. GPC with Waters 150C GPC was used to determine $M_n$, $M_w$, and Đ of the polymer products against a series of

monodisperse polystyrenes as standards in 1,2,4-trichlorobenzene (HPLC grade) at 120 °C. For multiple column usage, three GPC columns for high-temperature were connected in series of GPC-HT-803, -804, and -805 (product code: F62085700, F6208710, and F6208720, column size: 8.0 × 300 mm, usable temperature: 100 to 150 °C, particle size: 13 μm, pore size: 500, 1500, and 5000 Å, and, exclusion limit: 70, 400, and 40,000 kDa, Shodex$^{TM}$). The monodisperse polystyrene standards with a broad range of $M_w$s were used for the GPC calibration curves with the equation, $Y$ (log$M_w$) = −0.31854*X (min) + 9.02902 and the correlation factor, 0.9998350 (see the Supplementary Table 2 and Fig. 2). To avoid errors caused by extrapolation, Fig. 3 was replotted with the calibration curve and points, as shown in the Supplementary Fig. 5. Cyclic voltammograms of entries 17 and 18 on a platinum working electrode were obtained by AMETEK Versa STAT 3 at a scan rate of 100 mV s$^{-1}$. For a three-electrode cell system in a nitrogen bubbled 0.1 M tetra-$n$-butylammonium hexafluorophosphate ($n$-Bu$_4$NPF$_6$) solution in acetonitrile, Ag/Ag$^+$ electrode and a platinum wire were used as the reference electrode and counter electrode, respectively. A Fc/Fc$^+$ redox couple as an internal standard was used to calibrate the Ag/Ag$^+$ reference electrode. Oxidation potential of Fc/Fc$^+$ was set at −4.8 eV with respect to a zero vacuum level. $E_{HOMO}/E_{LUMO} = -(\varphi_{ox}^{onset}/\varphi_{red}^{onset} - \varphi_{ferrocene}^{onset} + 4.8)$ (eV) was used for calculating the HOMO/LUMO energy levels. AFM investigations were carried out by a multimode V microscope (Veeco, USA) in the tapping mode with a nanoscope controller using Si tips (Bruker). The AFM phase images were provided in the Supplementary Fig. 4. TEM analysis was performed using a JEOL USA JEM-2100F (Cs corrector) transmission electron microscope. GIWAXD measurement was carried out at the PLS-II 6D UNIST-PAL beamline of the Pohang Accelerator Laboratory in Korea. A double crystal monochromator with a 2D CCD detector (Rayonix SX165) was used to monochromate the X-ray coming from the in-vacuum undulator ($\lambda$ = 1.1099 Å), which was focused both horizontally and vertically ((450 (H) × 60 (V) μm$^2$ in FWHM

(full width at half maximum) @sample position) using K-B type mirrors. The 7-axis motorized stage was used for the fine alignment of samples, and the incidence angle of X-ray beam was set to be 0.12º for the neat polymers and the blend films. Diffraction angles were calibrated using a sucrose standard (Monoclinic, P21, $a = 10.8631$ Å, $b = 8.7044$ Å, $c = 7.7624$ Å, $\beta = 102.938°$) and the detector was located at a distance of ≈232 mm from the sample center.

**Device fabrication and characterizations.** The conventional devices were adopted to assess the performance of polymers:$PC_{71}BM$-based solar cells with the device architectures of glass/ITO/PEDOT:PSS/photoactive materials/Al. ITO-coated glasses were washed using deionized water, acetone and isopropyl alcohol, dried in an oven. Solutions of the PTB7-based polymers in chlorobenzene/1,8-diiodooctane solvent (97:3 vol%) at concentrations of 12 mg mL$^{-1}$ were used for the blending solutions polymer:$PC_{71}BM$ (1:1.7 wt%). PEDOT:PSS was spin-coated on the substrates at 4000 rpm for 1 min and dried at 150 °C for 10 min. The prepared blend solutions were spin-coated at 900 rpm for 2 min onto PEDOT:PSS-coated ITO in the glove box under nitrogen condition. Subsequently, the Al (100 nm) counter-electrode was thermally evaporated under a vacuum (B106 Torr), which defines the device area of 13 mm$^2$. $J$–$V$ characteristics were obtained in the glove box with a xenon arc lamp solar simulator under AM 1.5 G illumination (100 mW cm$^{-2}$) using a Keithley 2635 A source measurement unit, and QE system (Model QEX7) from PV measurements Inc. (Boulder, Colorado) was used for EQE measurement under ambient conditions.

**Data availability.** All relevant data supporting the findings of this study are available from the authors on request.

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

## Acknowledgements

This work was supported by the National Research Foundation of Korea (NRF) grant funded by the Korea government (MSIP) (2015R1A2A1A10053397 and 2015R1D1A1A0105791). GIWAXD measurements at PLS-II 6D UNIST-PAL beamline and 9A beamline were supported in part by MEST, POSTECH, and UNIST UCRF. We thank J. Lee, S. -H. Kang, and B. Lee at UNIST for providing the monomers for various other polymers.

## Author contributions

C.Y. and H.P. conceptualized the project. S.M.L. synthesized and characterized all the materials. K.H.P. and S.J. fabricated and tested the PSCs. S.M.L., H.P., and C.Y. developed the interpretation of the data and wrote the manuscript. The projects were supervised by C.Y. and H.P.
