## [Peer Review File(PDF 2345 kb) · Nature Communications]

COMMENTS TO AUTHOR:

Reviewer #1:

The paper addresses one of the main open questions (bottleneck-like) in the development of reliable organic photovoltaic technology, namely batch-to-batch variations in the device performance. I agree with the author's suggestion that it can be due to batch-to-batch variation in molecular weight (M_w) and polydispersity index (PDI) for a given conjugated polymer. The paper describes an elegant novel method (a stepwise heating protocol in the Stille polycondensation in conjunction with optimized processing) to obtain an ultrahigh-quality PTB7 polymer having high M_w and very narrow PDI. The suggestion has been directly confirmed: the resulting ultrahigh-quality polymer-based solar cells demonstrate up to 9.97% power conversion efficiencies (PCEs), which is an ~24% enhancement from the control device fabricated with commercially available PTB7. It is even more important that almost negligible batch-to-batch variations in the overall PCE values from ultrahigh-quality polymer-based devices is in evidence. I would agree with the authors' conclusion that the proposed stepwise polymerization can open a route for synthesizing high-quality semiconducting polymers that can significantly improve device yield for effective polymer-based PV and polymer-based electronics in general. Thus, there is no doubt about a novelty and importance of the demonstrated results. The paper is well and clearly written. In my view it is of interest to a wide community of readers of "Nature communications".

(1) It is clear that the demonstrated PCE optimization is due to increase in I_{sc} of the devices (Fig 4 and table 3). The authors did not discuss this effect. Though such discussion does not affect the main message of the paper (suggestion of the stepwise heating protocol) it may improve scientific quality of the manuscript. In this context I would recommend to discuss the results of recent paper by Ding, et al [J. Mater. Chem. A, 4, 7274, 2016]. The authors of that paper observed relationship between M_w of PTB7 polymer and I_{sc} and PCE of the PV cells of similar configuration. Decrease of the density of radical-type paramagnetic defects (detected by ESR) with M_w increase has been suggested as one of the possible underlying mechanisms.

[Response]

→ We appreciate the reviewer's valuable comments for our manuscript. As the reviewer pointed out, the effect on the solar cell performances should have been addressed with reasonable proof for increase in I_{sc} of the devices. There are many investigations which address the relationship between molecular weights of polymers and photovoltaic parameters, including the ESR characterizations for the density of radical defects of polymers, which the reviewer suggested in the paper published in *J. Mater. Chem. A*. Among the various characterizations, we found the increase of currents originated from the enhanced absorption depending on the molecular weights as evidenced by the UV-vis spectrometer. Hence, we plotted the non-normalized UV-vis spectra with the absorption coefficients, followed by the modified details in the revised manuscript. In addition, we cited the suggested paper with the additional statements, hoping to help the readers for broad understanding of the polymeric effects on the photovoltaic performance.

→ **Modified statements for Fig. 2**

Several reports have demonstrated the influence of polymer M_w s on various optical, physico-chemical, and morphological properties³⁶⁻⁴¹. In this regard, we explored the optical and electrochemical properties, morphology, and molecular organization for two representative batches (entries 17 and 18) using UV-vis absorption spectroscopy, cyclic voltammetry, atomic force microscopy, transmission electron microscopy, and grazing-incidence wide-angle X-ray diffraction (GIWAXD). The film absorption coefficients (ϵ) varied from 93,900 to 133,000 cm^{-1} for entries 17 and 18, implying that the increase of M_w s can result in the enhanced light-absorbing capability. Nonetheless, the two test samples exhibited similar physical features (nearly identical absorption shape, optical bandgap of 1.67 eV, HOMO:LUMO of -5.34:-3.61 eV, small surface roughness of ~1.0 nm, lamellar distance of ~19.5 Å, and π - π stacking distance of ~3.9 Å), as revealed by a series of the aforementioned analysis (Fig. 2), which to some extent may reflect the sufficiently high quality that correlates with nearly identical structural aspects.

→ **Modified statements for Fig. 4**

The increase in J_{SC} values is in good agreement with the results of external quantum efficiency (EQE) measurements from the corresponding devices (Fig. 4b). In addition, the trend of the EQE enhancements is well-matched with the increased absorption spectra of the blend films (Supplementary Fig. 5), which is consistent with what was previously reported in that the photocurrent correspondingly increases with M_w values^{47,48}. Consequently, we achieved a PCE of up to 9.97% ($PCE_{avg} = 9.69\%$) in the best-quality entry 18-based device, which, to our best knowledge, is the highest value reported for PTB7-based PSCs. More importantly, the entry 18-based device shows superior reproducibility in regard to all photovoltaic parameters; more than 30 devices lie within trivial error ranges of 5.88% (~95% confidence level), whereas the control devices (entry 19 and Commer-1) show large performance variations in each device batch (Fig. 4c). The origin of improvements in both the PCEs and reproducibility of PSCs based on entry 18 is consistent with what was previously suggested in that a lower density of radical-type paramagnetic defects in PTB7 with high M_w leads to improved device functionality⁴⁹. These observations imply that polymer chain quality control is an important contributor to resolving the challenging issue of device-to-device variations in PSCs.

→ **Added references in the main text**

47. Liu, C., *et al.* Molecular weight effect on the efficiency of polymer solar cells. *ACS Appl. Mater. Inter.* **5**, 12163-12167 (2013).

48. Intemann, J. J., *et al.* Molecular weight effect on the absorption, charge carrier mobility, and photovoltaic performance of an indacenodiselenophene-based ladder-type polymer. *Chem. Mater.* **25**, 3188-3195 (2013).

49. Ding, Z., *et al.* Efficient solar cells are more stable: the impact of polymer molecular weight on performance of organic photovoltaics. *J. Mater. Chem. A* **4**, 7274-7280 (2016).

→ **Modified figures in the main text**

Figure 2 | Optical, electrochemical, and morphological properties of entries 17 and 18. a, UV-vis absorption spectra and cyclic voltammograms. **b,** AFM height images in a scale of 500 nm with inset TEM images (5 nm scale). **c,** GIWAXD images of in-plane and out-of-plane.

→ **Modified tables in the Supplementary Information**

Supplementary Table 5. The optical and electrical properties of entries 17 and 18.

Polymers	$\lambda_{\max}^{\text{film}}$ (nm) [*]	$\epsilon_{\text{abs}}^{\text{film}}$ (cm ⁻¹) [†]	$E_{\text{g}}^{\text{opt}}$ (eV) [‡]	E_{HOMO} (eV) [§]	E_{LUMO} (eV) [§]	$E_{\text{g}}^{\text{elec}}$ (eV)
entry 17	627, 679	93,900	1.68	-5.34	-3.61	1.73
entry 18	627, 679	133,000	1.67	-5.34	-3.61	1.73

*Polymer films were spin-coated on glass substrates. †The absorption coefficient (ϵ) was obtained using the equation ($\epsilon = 2.303 \times (A/l)$), where A is the absorbance and l is the film thickness (in cm). ‡The optical band gap was determined from the onset of the UV-vis absorption spectra in the polymer films. §Energy levels are measured by cyclic voltammetry with Fc/Fc⁺ ($E_{\text{HOMO}} = -4.80$ eV) as the internal reference. ||The electronic band gap was calculated from $E_{\text{g}}^{\text{elec}} = E_{\text{LUMO}} - E_{\text{HOMO}}$.

(2) I would suggest to add ‘polymer molecular weight’ and ‘polydispersity index’ to key-word list.

[Response]

→ Thanks for careful suggestions. We added two primary words into the key-word list.

→ **Added words into the key-word list**

Keywords: batch-to-batch variations, polymer molecular weight, polydispersity index, preferential step-growth, stepwise Stille polycondensation

Reviewer #2:

The paper reports a modification to the synthesis of a typical Stille polymer, in which a low temperature ‘annealing’ step is used during the polymerization technique to improve the quality of the polymer (high molecular weight and lower dispersity). The paper will be of interest to the synthetic polymer community, but there are a number of issues to address before it could be considered for publication.

(1) Firstly, for the solar cell device performance, I don’t think the comparison with just the commercial material is a fair one, since the monomers used here have been extensively purified. A comparison with polymer made using standard conditions should be shown, at the same catalyst loadings, such that catalyst based impurities can be ruled out. This is essential in my view to enable a fair comparison. We do not know how the commercial polymer were prepared. For the actual devices, the thickness for each device should be given. The differences could just be related to different thicknesses. Please given non-normalized UV for each device to prove similar thickness.

[Response]

→ We thank the reviewer for careful and constructive comments. A fair comparison of photovoltaic performances should be made with an entry prepared by the conventional Stille polycondensation to eliminate the issues of monomer/catalyst impurities and others. Therefore, as the reviewer’s suggestion, we synthesized an additional sample as ‘**new entry 19**’ under the conventional Stille conditions with optimal 1.0 mol% Pd catalyst loading, followed by the detailed description of polymeric parameters in the revised manuscript. In addition, the solar cell performances including $J-V$ characteristics and EQE spectra were also included with variation tests in the photovoltaic parameters.

To properly address the quality effects of the polymeric materials on photovoltaic performances, as the reviewer pointed out, the values of thickness of each device should be provided to focus solely on M_w and PDI influences. We measured the thickness of the actual cells to show similar values in four different types of devices and added the results into the footnotes of table. Moreover, non-normalized UV-vis spectra of pristine and blending samples with PC₇₁BM were plotted with the absorption coefficients. Following the optical absorptivity from UV-vis measurement by excluding the thickness issues, the trends of as-observed photovoltaic performances, especially J_{SC} values are

clearly comprehensible from entry 17 to commercial purchase.

(We provided the revised results for this first issue combined with the second comment.)

(2) Secondly, in order to be suitable for this journal the wide scope of this approach should be shown. Currently only one polymer system is considered. What is the influence on different monomeric system (for example DPP containing polymers which the authors work on, or thiophene containing systems)? The scope has to be broadened – one polymer has limited interest.

[Response]

→ We are very grateful for the reviewer's major concerns. It is totally agreed that the scope of stepwise Stille polymerization in our submitted work was limited to only one polymeric system. To further explore the versatility of our stepwise polycondensation, we prepared four polymers with different monomers from PTB7 polymer, which are representatively adopted in the electronic applications. The synthetic routes and optimization conditions for each polymer varied from catalyst system to solvent, were added into the detailed experimental sections in the revised Supplementary Information. In addition to the comparison of conventional and stepwise methods with GPC data for our selected polymers, the photovoltaic devices from both kinds of the batches were comparatively investigated, resulting in improved PCEs in the stepwise batches, which verifies that our methodology is a universal and simple yet efficient tool for ultrahigh-quality semiconducting precision.

→ **Modified statements in p.8**

Through further optimization processes, we obtained the best-quality PTB7 polymer having $M_w = 223$ kDa and PDI = 1.21 using a combination of 0.026 M concentration and a catalyst concentration of 1.0 mol%. For a fair comparison, the conventional Stille polycondensation was also performed using the optimized condition above, producing PTB7 with only moderate M_w (entry 19), which clearly corroborates the efficacy of stepwise polymerization to achieve a higher-quality PTB7 with high M_w and narrow PDI. Supplementary Table 3 lists the remainder of the conditions tested over the course of our study

→ **Modified statements in p.10**

Next, we further compared and quantified the difference between the resulting polymers prepared from our optimized stepwise and conventional protocols (entries 17, 18, and 19) and commercially available PTB7 products, which we purchased from Derthon and 1-Material companies. As shown in Fig. 3, both entries 17 and 18 show significantly higher M_w and narrower PDI values than those of the entry 19 and commercial products, which we determined by the same processing GPC analysis (Supplementary Table 6). Note that the different commercial products used in this study are denoted as Commer-1 and Commer-2, respectively. These results clearly indicate that stepwise polymerization is an effective strategy to achieve a narrow distribution of high- M_w polymers.

→ **Modified statements in p.11**

To assess the effect of polymer quality control on photovoltaic properties, we fabricated archetypal single junction PSC devices (ITO/PEDOT:PSS/PTB7:PC₇₁BM/Al) based on four polymers (entries 17, 18, 19, and Commer-1). In accordance with the aforementioned optimized processing conditions, we spin-coated the PTB7:PC₇₁BM blend (1:1.7 wt%), 12 mg mL⁻¹ concentration, in a mixed solvent of chlorobenzene (CB):1,8-diodooctane (DIO) (97:3 vol%). We chose Commer-1 as a commercial reference sample because of its higher quality from a materials standpoint and better PSC characteristics in the initial screening test relative to that of Commer-2 (Supplementary Fig. 4). The result of entry 19 obtained from conventional Stille polymerization was also included in the evaluation of the PSC performance for the sake of comparison.

Fig. 4a shows the current density–voltage (J – V) curves measured under the simulated AM 1.5G at 100 mW cm⁻² irradiance, and Table 3 shows the corresponding key photovoltaic parameters. The Commer-1:PC₇₁BM control device showed a short-circuit current density (J_{sc}) of 16.5 mA cm⁻², an open-circuit voltage (V_{oc}) of 0.72 V, and a fill factor (FF) of 61%. This indicates a PCE of 8.02% (PCE_{avg} = 7.26%), which is similar to the results previously reported from the highly optimized PTB7-based devices⁴²⁻⁴⁶. The device performance for the entry 19 was lowest with a PCE of 6.98%

($PCE_{avg} = 5.92\%$). On the other hand, both entries 17- and 18-based devices display significant improvements in the J_{SC} , having similar V_{OC} s of 0.71 V and FF s of 62%. The increase in J_{SC} values is in good agreement with the results of external quantum efficiency (EQE) measurements from the corresponding devices (Fig. 4b). In addition, the trend of the EQE enhancements is well-matched with the increased absorption spectra of the blend films (Supplementary Fig. 5), which is consistent with what was previously reported in that the photocurrent correspondingly increases with M_w values^{47,48}. Consequently, we achieved a PCE of up to 9.97% ($PCE_{avg} = 9.69\%$) in the best-quality entry 18-based device, which, to our best knowledge, is the highest value reported for PTB7-based PSCs. More importantly, the entry 18-based device shows superior reproducibility in regard to all photovoltaic parameters; more than 30 devices lie within trivial error ranges of 5.88% (~95% confidence level), whereas the control devices (entry 19 and Commer-1) show large performance variations in each device type (Fig. 4c). The origin of improvements in both the PCEs and reproducibility of PSCs based on entry 18 is consistent with what was previously suggested in that a lower density of radical-type paramagnetic defects in PTB7 with high M_n leads to improved device functionality⁴⁹. These observations imply that polymer chain quality control is an important contributor to resolving the challenging issue of device-to-device variations in PSCs.

Moreover, to explore the versatility of the stepwise heating protocol, we attempted to prepare various other polymers, including thienopyrroledione-, diketopyrrolopyrrole-, quinoxaline-, and thiophene-based monomeric systems by using conventional- and stepwise-heating systems, respectively (Supplementary Scheme 1). As can be seen in the relevant GPC results (Supplementary Fig. 6 and Table 8), in all cases, the stepwise polymerization produced the polymeric components with higher M_w and narrower PDI values compared to those of the corresponding ones obtained from the conventional method. The PSCs based on the resulting polymers were comparatively investigated in the same device architecture as described above (Supplementary, details of the optimized conditions for each polymer). The devices based on the stepwise batches showed higher PCEs than those of the corresponding ones fabricated from the conventional method (Supplementary Fig. 7). Note that the higher-quality PBDT-TPD prepared from the stepwise tool exhibited a slightly lower PCE than that of the sample with conventional tool, due to its poorer film formation with aggressive aggregation behavior induced by the low solubility and processability upon the device fabrication^{50,51}. These overall results propose the feasibility of stepwise Stille polymerization for a broader application.

➔ **Added references in the main text**

46. Park, K. H., An, Y., Jung, S., Park, H. & Yang, C. Locking-in optimal nanoscale structure induced by naphthalenediimide-based polymeric additive enables efficient and stable inverted polymer solar cells. *ACS Nano* **11**, 7409-7415 (2017).

47. Liu, C., *et al.* Molecular weight effect on the efficiency of polymer solar cells. *ACS Appl. Mater. Inter.* **5**, 12163-12167 (2013).

48. Intemann, J. J., *et al.* Molecular weight effect on the absorption, charge carrier mobility, and photovoltaic performance of an indacenodiselenophene-based ladder-type polymer. *Chem. Mater.* **25**, 3188-3195 (2013).

49. Ding, Z., *et al.* Efficient solar cells are more stable: the impact of polymer molecular weight on performance of organic photovoltaics. *J. Mater. Chem. A* **4**, 7274-7280 (2016).

50. Kingsley, J. W., *et al.* Molecular weight dependent vertical composition profiles of PCDTBT:PC₇₁BM blends for organic photovoltaics. *Sci. Rep.* **4**, 5286 (2014).

51. Subbiah, J., *et al.* Organic solar cells using a high-molecular-weight benzodithiophene-benzothiadiazole copolymer with an efficiency of 9.4%. *Adv. Mater.* **27**, 702-705 (2015).

→ Added figures and tables in the main text

Table 2 | Stepwise Stille polymerization with the optimization data.

Entry	Solvent (Conc.)	Catalyst (mol%)	M_n (kDa) [*]	M_w (kDa) [*]	PDI [*]	Yield (%) [†]
11	toluene/DMF (0.05 M)	Pd(PPh ₃) ₄ (4.0)	67.2	90.1	1.34	- [‡]
12 [§]	toluene/DMF (0.05 M)	Pd(PPh ₃) ₄ (4.0)	43.9	82.6	1.88	- [‡]
13 [§]	toluene/DMF (0.05 M)	Pd(PPh ₃) ₄ (4.0)	42.5	99.8	2.35	- [‡]
14	toluene/DMF (0.10 M)	Pd(PPh ₃) ₄ (4.0)	31.3	65.1	2.08	63
15	toluene/DMF (0.026 M)	Pd(PPh ₃) ₄ (4.0)	82.1	125	1.52	65
16	toluene/DMF (0.026 M)	Pd(PPh ₃) ₄ (3.0)	110	151	1.37	81
17	toluene/DMF (0.026 M)	Pd(PPh ₃) ₄ (2.0)	151	195	1.29	79
18	toluene/DMF (0.026 M)	Pd(PPh ₃) ₄ (1.0)	184	223	1.21	85
19	Conventional Polymerization		46.9	75.8	1.62	72

The stepwise Stille polycondensation was carried out under an argon atmosphere in a long Schlenk tube of monomer **1**, **2** and Pd(PPh₃)₄ and procedures included the initial heating at 120 °C for 1 h, the cooling step at 60 °C for 11 h, and the final heating at 120 °C for 1 day. ^{*} M_n , M_w , and PDI values were determined from GPC measurement using 1,2,4-trichlorobenzene at 120 °C calibrated with polystyrene as standard. [†]Yields were estimated from the amounts of the chloroform fractions. [‡]Each fraction was extracted by a syringe, then precipitated in methanol with Soxhlet purification for only GPC analysis. [§]The cooling temperature was investigated at 80 °C and 100 °C for entries 12 and 13, respectively. ^{||}1.0 mol% Pd catalyst was employed for conventional Stille polymerization.

Figure 3 | GPC data for test batches and commercial polymers. GPC profiles of entries 17, 18, 19, Commer-1, and -2. The multiple curves were deconvoluted through the fitting of each GPC curve.

Figure 4 | Photovoltaic performances of entries 17, 18, and 19 with comparison of Commer-1. a, J - V characteristics for BHJ solar cells. b, EQE spectra in a range from 300 to 900 nm. c, Variations in device parameters of J_{SC} , V_{OC} , and PCEs.

Table 3 | Device parameters for entries 17, 18, 19, and Commer-1.

Samples	J_{SC} (mA cm ⁻²)	V_{OC} (V)	FF (%)	PCE (%) max/avg.*
entry 17 [†]	19.8±0.4	0.71±0.02	62±2	9.35/8.93*
entry 18 [†]	20.9±0.3	0.71±0.01	62±2	9.97/9.69*
entry 19 [†]	16.3±0.6	0.72±0.03	60±3	6.98/5.92*
Commer-1 [†]	16.5±0.5	0.72±0.02	61±1	8.02/7.26*

*The average values were obtained from over 30 devices with the standard deviation. [†]The thicknesses of the polymer blending films with PC₇₁BM are 85, 80, 78, and 83 nm, respectively.

➔ Added figures in the Supplementary Information

Supplementary Figure 5. UV-vis spectra of the blending films with PC₇₁BM (1:1.7 wt%).

→ **Added experimental sections in Supplementary Information**

Stepwise Stille polycondensation of poly[4,8-bis((2-ethylhexyl)oxy)benzo[1,2-b:4,5-b']dithiophene-2,6-diyl-alt-5-octylthieno[3,4-c]pyrrole-4,6-dione-1,3-diyl] (PBDT-TPD): To a mixture of monomers **1** (100 mg, 0.13 mmol) and **3** (54.8 mg, 0.13 mmol) in a binary mixture of toluene and DMF (0.10 M) as a 4:1 ratio, the solution of Pd(PPh₃)₄ in toluene (2.6 mM) was injected with subsequent argon purging for 30 min. The long Schlenk tube was placed in a preheated oil bath at 120 °C for 1 h for the initial stage, after which allowed the temperature to reduce and kept the tube cooled for 11 h. The reaction temperature was set again to 120 °C and continuously maintained for 1 day. Cooling down to room temperature, the precipitates in methanol was purified and the fraction from chloroform was characterized under the same process conditions used for the PTB7.

Stepwise Stille polycondensation of poly[2,5-bis(2-octyldodecyl)pyrrolo[3,4-c]pyrrole-1,4(2H,5H)-dione-3,6-diyl-alt-2,2':5',2'':5'',2'''-quaterthiophene-5,5'''-diyl] (PDPP-biTh): To a mixture of monomers **4** (48.3 mg, 0.098 mmol) and **5** (100 mg, 0.098 mmol) in toluene (0.078 M), the solution of Pd(PPh₃)₄ in toluene (2.0 mM) was injected with subsequent argon purging for 30 min. The long Schlenk tube was placed in a preheated oil bath at 100 °C during color change to green for the initial stage, after which allowed the temperature to reduce and kept the tube cooled for 11 h. The reaction temperature was set again to 100 °C and continuously maintained for 1 day. Cooling down to room temperature, the precipitates in methanol was purified and the fraction from chloroform was characterized under the same process conditions used for the PTB7.

Stepwise Stille polycondensation of poly[6-fluoro-2,3-bis-(3-octyloxyphenyl)quinoxaline-5,8-diyl-alt-thiophene-2,5-diyl] (PFQx-Th): To a mixture of monomers **6** (57.3 mg, 0.14 mmol) and **7** (100 mg, 0.14 mmol) in chlorobenzene (0.11 M), Pd₂(dba)₃/P(*o*-tolyl)₃ (1.4/5.6 μmol) was added with subsequent argon purging for 30 min. The long Schlenk tube was placed in a preheated oil bath at 140 °C for 2 h for the initial stage, after which allowed the temperature to reduce and kept the tube cooled for 10 h. The reaction temperature was set again to 140 °C and continuously maintained for 2 days. Cooling down to room temperature, the precipitates in methanol was purified and the fraction from chloroform was characterized under the same process conditions used for the PTB7.

Stepwise Stille polycondensation of poly[5-(2-hexyldodecyl)thieno[3,4-c]pyrrole-4,6-dione-1,3-diyl-alt-4',4''-didodecyl-2,2':5',2''-terthiophene-5,5'''-diyl] (PTPD-Th): To a mixture of monomers **6** (39.5 mg, 0.097 mmol) and **8** (100 mg, 0.097 mmol) in a binary mixture of toluene and DMF (0.039 M) as a 4:1 ratio, Pd₂(dba)₃/P(*o*-tolyl)₃ (1.4/5.6 μmol) was added with subsequent argon purging for 30 min. The long Schlenk tube was placed in a preheated oil bath at 120 °C for 1 h for the initial stage, after which allowed the temperature to reduce and kept the tube cooled for 11 h. The reaction temperature was set again to 120 °C and continuously maintained for 1 day. Cooling down to room temperature, the precipitates in methanol was purified and the fraction from chloroform was characterized under the same process conditions used for the PTB7.

Device fabrication of PBDT-TPD, PDPP-biTh, PFQx-Th, and PTPD-Th: The devices based on PBDT-TPD, PDPP-biTh, PFQx-Th, and PTPD-T polymers were fabricated under the similar process conditions used for the PTB7-based devices, except for the followings: chlorobenzene/1,8-diiodooctane solvent (97:3 vol%, 8 mg mL⁻¹ concentration), polymer:PC₇₁BM (1:1.5 wt%) blend ratio, and spin-coating at 1500 rpm for PBDT-TPD; dichlorobenzene/chloroform solvent (1:4 vol%,

8 mg mL⁻¹ concentration), polymer:PC₇₁BM (1:1.5 wt%) blend ratio, and spin-coating at 3000 rpm for PDPP-biTh; dichlorobenzene/chloroform solvent (4:1 vol%, 8 mg mL⁻¹ concentration), polymer:PC₇₁BM (1:1.5 wt%) blend ratio, and spin-coating at 1000 rpm for PFQx-Th; chloroform/1,8-diiodooctane solvent (98:2 vol%, 10 mg mL⁻¹ concentration), polymer:PC₇₁BM (1:2 wt%) blend ratio, spin-coating at 3000 rpm for 1 min, and annealing at 120 °C for 10 min for PTPD-Th, respectively.

➔ Added figures and table in Supplementary Information

Supplementary Scheme 1. Synthetic routes for PBDT-TPD (a), PDPP-biTh (b), PFQx-Th (c), and PTPD-Th (d).

Supplementary Figure 6. Comparison of stepwise and conventional Stille polycondensations by GPC profiles and structures for PBDDT-TPD (a), PDPP-biTh (b), PFQx-Th (c), and PTPD-Th (d).

Supplementary Table 8. The results of GPC measurements for additional four polymers synthesized by stepwise and conventional methods.

Polymers	Temp. (°C)	Time (h)	M_n (kDa) [*]	M_w (kDa) [*]	PDI [*]	Yield (%) [†]
PBDDT-TPD	120	1	8.21	18.1	2.19	- [‡]
	60	11	23.2	43.1	1.87	- [‡]
	120	24	91.3	127	1.40	73
	Conventional Polymerizations ⁴		30.1	64.2	2.14	82
PDPP-biTh	100	- [§]	11.4	34.4	3.02	- [‡]
	60	11	27.4	59.6	2.17	- [‡]
	100	24	81.2	101	1.26	88
	Conventional Polymerizations ⁵		37.7	71.8	1.91	85
PFQx-Th	140	2	12.5	29.9	2.40	- [‡]
	60	10	33.9	71.4	2.11	- [‡]
	140	48	82.1	112	1.36	54
	Conventional Polymerizations ⁶		27.3	51.5	1.88	69
PTPD-Th	120	1	8.28	11.8	1.43	- [‡]
	60	11	21.2	34.0	1.60	- [‡]
	120	24	98.6	139	1.41	89
	Conventional Polymerizations ⁷		40.5	74.8	1.85	57

The conventional batches were carried out following the cited papers. ^{*} M_n , M_w , and PDI values were determined from GPC measurement using 1,2,4-trichlorobenzene at 120 °C calibrated with polystyrene as standard. [†]Yields were estimated from the amounts of the chloroform fractions. [‡]Each fraction was extracted by a syringe, then precipitated in methanol with Soxhlet purification for only GPC analysis. [§]In the optimized conditions for PDPP-biTh case, the initial step was proceeded until the solution color was changed to green.

Supplementary Figure 7. Comparison of J - V characteristics with stepwise and conventional batches for PBDT-TPD (a), PDPP-biTh (b), PFQx-Th (c), and PTPD-Th (d).

➔ **Added references in the Supplementary Information**

- Zhang, Y., *et al.* Efficient polymer solar cells based on the copolymers of benzodithiophene and thienopyrroledione. *Chem. Mater.* **22**, 2696-2698 (2010).
- Liu, F., *et al.* Efficient polymer solar cells based on a low bandgap semi-crystalline DPP polymer-PCBM blends. *Adv. Mater.* **24**, 3947-3951 (2012).
- Dutta, G. K., *et al.* Synthesis of fluorinated analogues of a practical polymer TQ for improved open-circuit voltages in polymer solar cells. *Polym. Chem.* **5**, 2540-2547 (2014).
- Guo, X., *et al.* Polymer solar cells with enhanced fill factors. *Nat. Photonics* **7**, 825-833 (2013).

(3) The authors state the improvement is due to the fact that at 60C, ‘the low-Mw species would react preferentially, and thus, the window of the Mw distributions can potentially be reduced’ (page 7). I think some evidence should be given for this statement – why is the low MW species more reactive. I would like to see the actual GPC traces shown to prove this (rather than tabulated data). Is the polymer all soluble at this temperature? Is some high MW material simply precipitating, and then redissolves at higher temperature.

[Response]

- ➔ We are thankful for the reviewer’s kind comments. Rather than the tabulated data for the entries optimized with the stepwise heating protocols, GPC plots would be powerful to show that low- M_w polymeric chains are grown selectively during low-temperature maturing step. Moreover, precipitating samples from entry S9 were all soluble in the reacting binary solvents at room

temperature. We added GPC traces and corresponding vial pictures into the Supplementary Information.

→ **Modified statements in p.7**

We first performed polymerization at the optimal temperature (120 °C) for 1 h and then cooled the reaction mixture to 60 °C for 11 h as a selectivity control step. Over this time period, the low- M_w species would react preferentially, and thus, the window of the M_w distributions can potentially be reduced (Supplementary Fig. 2b). It is noteworthy that all of the polymeric fractions formed at the low-temperature step (60 °C) are readily dissolved in the mixed reaction solvent (Supplementary Fig. 2c), suggesting that the reactivity variation caused by their solubility issues is negligible. After the cooling stage, we reverted the reaction temperature to the initial setting for finalizing the step-growth reaction.

→ **Added figures in the Supplementary Information**

Supplementary Figure 2. GPC profiles of time-dependent growing trends of polymeric chains in conventional Stille polycondensation for entries 1 and 10 (a) and in the stepwise method for entry 11 (b) with pictures of a set of three steps clearly dissolved in toluene/DMF for entry S9 (c).

(4) The high hazards associated with organotin, particularly trimethyltin, should be highlighted in the experimental.

[Response]

→ We appreciate the reviewer's comments for high toxicity of organotin compounds and totally agree further notification of their hazards. For following researchers to synthesize monomers with tin-related functional groups, we made toxic properties of organotin reagents highlighted in the Supplementary Information with the reported papers.

→ **Added statements in the Supplementary Information**

Materials and Instruments: All the chemicals and reagents were bought from Sigma-Aldrich, Alfa Aesar chemical company, Tokyo Chemical Industry Co., Ltd., Derthon Optoelectronic Materials Science Technology Co., Ltd. and unless otherwise specified used without any further purification. (*Caution:* Trimethyltin chloride and other organotin-related compounds are highly toxic to cause irritation and burns of the skin and eyes with fatal damage on the central nervous system. High and repeated exposure can cause loss of hearing, weakness, confusion, and seizures^{1,2}.) 2,6-Bis(trimethyltin)-4,8-bis(2-ethylhexyloxy)benzo[1,2-*b*:4,5-*b'*]dithiophene was synthesized according to the literature³ and 2-ethylhexyl-4,6-dibromothiopheno[3,4-*b*]thiophene-2-carboxylate was purchased from Derthon, both of which were purified with recrystallization and chromatography using a reverse-phase column, JAIGEL-ODS-AP, SP-120-10 (2.0 cmφ × 25 cm, i.d., Japan Analytical Industry Co., Ltd.) in a mixture of acetonitrile and THF as the mobile phase.

➔ **Added references in the Supplementary Information**

1. Z. Liu, G. L., *et al.* Study on acute toxicity and changes of serum ions in rats, mice and rabbits treated with trimethyltin chloride. *Chin. Occup. Med.* **35**, 197 - 199 (2008).
2. Chen, J., *et al.* Trimethyltin chloride (TMT) neurobehavioral toxicity in embryonic zebrafish. *Neurotoxicol. Teratol.* **33**, 721-726 (2011).
3. Liang, Y., *et al.* Highly efficient solar cell polymers developed via fine-tuning of structural and electronic properties. *J. Am. Chem. Soc.* **131**, 7792-7799 (2009).

#Point-to-point responses of NCOMMS-17-19748B

COMMENTS TO AUTHOR:

Reviewer #1:

I am satisfied with the authors' response to my comments and to those of referee 2. The revised paper can be accepted as it is.

- We are deeply grateful for your acceptance of our revised manuscripts following the issues which you concerned in the first version.

Reviewer #2:

I think all the changes the authors have made have improved the paper, since they now show both greater scope and a direct comparison to the traditional Stille polymerisation. I am supportive of publication, but there are still 3 issues I would like to address:

- We really appreciate your previous constructive comments which led to the great improvement on our results. In this second revision, we tried our best efforts to make this work much acceptable for the broad readers, following your valuable insights.

(1) A minor point, but IUPAC now recommends the use of D bar (\bar{D}) rather than PDI. It would be good to update.

[Response]

- Thanks for your kind suggestion. We changed the abbreviation, PDI of polydispersity index to the D bar, \bar{D} in our manuscript.

(2) Secondly, to my eye some of the GPC traces seem very broad for the quoted PDI's in the tables. For example figure 2Sa, which corresponds to PDI's of 3.15, 3, and 2.2 (from bottom to top) and figure 2SB (entry 11) which is quoted to have a PDI of 1.34. The plot B seems much broader than I would expect for such a PDI, and rather similar to S2A (top). Similarly fig 3, entries 17 and 18 have a very long low weight tail for such narrow PDI's (1.20 and 1.21 from the table). Therefore please plot elugrams in weight distribution mode, rather than elution time. It is much clearer to see the distribution in this mode. The authors should also include in SI the actual elugrams with the calibration points marked such that the long weight tail can be seen to be included in the calculations.

[Response]

- We thank the reviewer for the careful and constructive comments. It is totally agreed that the elugrams for GPC data would be much clearer to compare their distribution in weight distribution mode. Hence, we additionally inserted the $\log M_w$ scales of GPC plots of Figure 3 into the Supplementary Information and changed the elugrams of Figure S2 in elution time to ones in $\log M_w$. In addition, the new figure with the calibration points and lines was added for the long weight tails of GPC traces in Figure 3. (To make the paper consistent with the revised manuscripts, we changed the sequence of supplementary figures and tables as highlighted by color shading.)

→ Modified Figure in the Supplementary Information

Supplementary Figure 3. GPC profiles of time-dependent growing trends of polymeric chains by $\log M_w$ scales in conventional Stille polycondensation for entries 1 and 10 (a) and in the stepwise method for entry 11 (b) with pictures of a set of three steps clearly dissolved in toluene/DMF for entry S9 (c).

→ Added Figure in the Supplementary Information

Supplementary Figure 5. GPC plots of Figure 3 in weight distribution mode (a) and with the calibration points and lines (b).

(3) Please include the GPC columns used (type and number) and the range of the calibration standards used. It would also be helpful to show (in SI) the calibration curve versus one of the elugrams.

[Response]

→ We totally agree that our GPC conditions will be helpful for following researchers to measure polymeric parameters for polymers. We added the information of our GPC columns with specific types and product numbers and the standards used for the calibration curves into the Supplementary Information. Furthermore, following the reviewer's valuable comment, the calibration curves were inserted with the correlation factors and the equation.

➔ **Added statements in the Supplementary Information**

Number-average (M_n) and weight-average (M_w) molecular weights, and polydispersity index (\mathcal{D}) of the polymer products were determined by gel permeation chromatography (GPC) with Waters 150C GPC using the columns for high-temperature measurement, GPC-HT-803, 804, and 805 (product code: F62085700, F6208710, and F6208720, usable temperature: 100 – 150 °C, particle size: 13 μm , pore size: 500, 1500, and 5,000 Å, and column size: 8.0 \times 300 mm, Shodex™) against a series of monodisperse polystyrene as standards in 1,2,4-trichlorobenzene (HPLC grade) at 120 °C. The monodisperse polystyrene standards with a broad range of M_w s were used for the GPC calibration curves with the equation, $Y(\log M_w) = -0.31825 * X(\text{min}) + 9.02758$ and the correlation factor, 0.9997783 (see the Supplementary Table 2 and Figure 2).

➔ **Added Figure in the Supplementary Information**

Supplementary Figure 2. GPC profiles of the broad ranges of the polystyrene standards and the calculated calibration points and line with the equation and the correlation factor.

➔ **Added Table in the Supplementary Information**

Supplementary Table 2. The standard groups of monodisperse polystyrenes used for GPC calibrations with their polymeric parameters.

Standards	M_w (kDa)	$\log M_w$	Elution Time _{max} (min)
Standard	217	5.34	11.65
Group 1	127	5.09	12.39
(Std G1)	20.1	4.30	14.85
Standard	282	5.45	11.17
Group 2	96.1	4.98	12.71
(Std G2)	6.31	3.80	16.41

#Point-to-point responses of NCOMMS-17-19748C

COMMENTS TO AUTHOR:

Reviewer #2:

I thank the authors for added the required information. I think the paper can be accepted now, after two minor issues are resolved.

- With our pleasure to your scientific concerns, we apologize for the previous responses to make you confused at all. For obvious descriptions of your issues, we modified the manuscript and supplementary one with our best efforts. If any doubts or confusion in the experimental data, please let us know to make this result comprehensible for the broad readers.

Polymer 18 - the best performing in terms of solar cells - clearly has a higher tail than the highest weight standard (see supplementary figure 5). They should just highlight it is above the calibration range, and therefore maybe inaccurate. Better of course would be to measure with a higher weight standard.

[Response]

- Thanks for the reviewer's totally reasonable comments. For reliability with molecular weight calculations, an actual region from start to end of integration should be in ranges of the known calibration curves. To avoid errors caused by extrapolation restricted to the calibration region, we measured the additional standard group including a high-molecular-weight monodisperse polystyrene standard denoted as Std G3 and still this group was highly well-matched with the previous calibration equation after few adjustments. Based on the adjusted calibration curves, we modified the figures and statements in the Supplementary Information.

→ Modified Statements in the Supplementary Information

The monodisperse polystyrene standards with a broad range of M_w s were used for the GPC calibration curves with the equation, $Y(\log M_w) = -0.31854 * X(\text{min}) + 9.02902$ and the correlation factor, 0.9998350 (see the Supplementary Table 2 and Figure 2).

→ Modified Figures in the Supplementary Information

Supplementary Figure 2. GPC profiles of the broad ranges of the polystyrene standards and the calculated calibration points and line with the equation and the correlation factor.

Supplementary Figure 5. GPC plots of Figure 3 in weight distribution mode (a) and with the calibration points and line (b).

→ **Modified Table in the Supplementary Information**

Supplementary Table 2. The standard groups of monodisperse polystyrenes used for GPC calibrations with their polymeric parameters.

Standards	M_w (kDa)	$\log M_w$	Elution Time _{max} (min)
Standard	217	5.34	11.65
Group 1	127	5.09	12.39
(Std G1)	20.1	4.30	14.85
Standard	282	5.45	11.17
Group 2	96.1	4.98	12.71
(Std G2)	6.31	3.80	16.41
Standard	508	5.71	10.43
Group 3	282	5.45	11.17
(Std G3)	32.9	4.52	14.13

The GPC equipment data given is not very clear. Are all three of these columns used? In series? Please state.

[Response]

→ We are deeply sorry for confusing with our description of GPC specifications. Typically, as the reviewer knows, the resolution of separation in GPC columns increases as the length of column increases. In order to make the separation efficient, it is a widely used and common method to connect multiple columns in series than use a single of long column. We adopted this multiple column usage of three Shodex™ high-temperature ones with different target molecular weight ranges. For clear description, we modified our experimental details in the Supplementary Information.

→ **Modified Statements in the Supplementary Information**

Number-average (M_n) and weight-average (M_w) molecular weights, and polydispersity index (\mathcal{D}) of the polymer products were determined by gel permeation chromatography (GPC) with Waters 150C GPC against a series of monodisperse polystyrenes as standards in 1,2,4-trichlorobenzene (HPLC grade) at 120 °C. For multiple column usage, three GPC columns for high-temperature were

connected in series of GPC-HT-803, -804, and -805 (product code: F62085700, F6208710, and F6208720, column size: 8.0×300 mm, usable temperature: 100 – 150 °C, particle size: 13 μm , pore size: 500, 1500, and 5,000 Å, and, exclusion limit: 70, 400, 4,000 kDa, Shodex™). The monodisperse polystyrene standards with a broad range of M_w s were used for the GPC calibration curves with the equation, $Y (\log M_w) = -0.31854 * X (\text{min}) + 9.02902$ and the correlation factor, 0.9998350 (see the Supplementary Table 2 and Figure 2).

#Point-to-point responses of NCOMMS-17-19748D

COMMENTS TO AUTHOR:

Reviewer #2:

All my minor comments have been addressed and I am happy to recommend acceptance.